# OpenReview forum: "Language Conditioned Traffic Generation"
_robot-learning.org/CoRL/2023/Conference — CoRL 2023 Poster_

### Official Review · Reviewer_5bXt · 2023-07-10

**Confidence:** 3
**Originality:** Very Good
**Technical Quality:** Very Good
**Clarity Of Presentation:** Very Good
**Impact:** 4

**Recommendation:**

Strong Accept: I recommend accepting the paper and will argue for my recommendation even if other reviewers hold a different opinion.

**Review:**

Strength

1.	The method that uses GPT-4 to generate traffic scenes is interesting and novel.
2.	The performance improvement is decent.
3.	The method can be used for language-conditioned traffic scene editing, which is never shown by previous methods before.
4.	The paper writes well

Weakness

1.	It would be interesting to include an ablation experiment to study whether open-sourced LLMs can replace GPT-4 in the system.


**Quality Of The Limitations Section:**

Limitations are addressed clearly

**Questions For Rebuttal:**

1.	Fig.3 is not referred to in the paper.
2.	The Large Language Model paragraph in the related work section should also cite GPT3 (when referring to in-context learning) and InstructGPT (when referring finetune with human feedback)
3.	Is the covariance matrix in L159 diagonal?
4.	In L164, what do you mean ‘sample’ the most probable values, is it greedy search?


**Robotics Focus:**

Highly relevant to robotics but no hardware experiments

**Summary Of Paper:**

This paper proposes a novel method to generate traffic scenes conditioned on languages. Their method contains three components, a GPT-4 based Interpreter that converts language into features, a Retrieval that retrieves suitable maps based on feature similarity and a Generator network that generates traffic agents based on maps and previous features. Experimental results show that their method outperforms previous baselines, and generates traffic scenes that are both more realistic and aligned with the language.

**Summary Of Recommendation:**

I think the method shows nice novelty and performance. In addition, I like the usage of language-conditioned traffic scene editing, it is quite novel. I think the method is potentially important for autonomous driving and deserves a presentation at the conference.

---

### Official Review · Reviewer_zVxc · 2023-07-11

**Confidence:** 3
**Originality:** Very Good
**Technical Quality:** Very Good
**Clarity Of Presentation:** Very Good
**Impact:** 4

**Recommendation:**

Weak Accept: I recommend accepting the paper, but will not argue for my recommendation if the majority of other reviewers have a different opinion.

**Review:**

This paper is well written and the idea is novel. Experimental results show much better results than baseline methods.

**Quality Of The Limitations Section:**

Limitations are addressed clearly

**Questions For Rebuttal:**

It will be interesting to show results w/ different LLMs and study how LLM can influence the results.

**Robotics Focus:**

Highly relevant to robotics but no hardware experiments

**Summary Of Paper:**

This paper proposes a method LCTGen that can leverage LLM to generate traffic scenes. It has 3 components: interpreter, generator and encoder. The interpreter convert natural language into a structured representation and then generator takes the structured representation and map to generate realistic traffic. Experimental results show that LCTGen can generate realistic and interesting traffic scenarios and this technique can be applied to applications such as instructional traffic scenario editing and controllable driving policy evaluation.

**Summary Of Recommendation:**

Using language to generate traffic scene is an interesting idea. It is much more efficient than collecting real traffic pattern or manually generate traffic scenes. I would recommend accepting this paper.

---

### Official Review · Reviewer_iJXw · 2023-07-13

**Confidence:** 4
**Originality:** Very Good
**Technical Quality:** Good
**Clarity Of Presentation:** Good
**Impact:** 4

**Recommendation:**

Weak Accept: I recommend accepting the paper, but will not argue for my recommendation if the majority of other reviewers have a different opinion.

**Review:**

Strengths:

1. Most of the paper is well written, and is easy to read.
2. The idea of leveraging LLMs for realizing, and controlling complex traffic scenarios is quite novel, and interesting.
It has immense potential to generate and augment dataset with rare and safety critical scenarios.
3. The baselines are evaluated on scene reconstruction, and language conditioned simulation, and the results are
backed up with human studies.

Weaknesses:

- While overall the paper is easy to read, there are certain sections which aren't easy to comprehend as follows:
 - Abstract and Introduction: These sections don't holistically provide the motivation, and future scope and applications of this work, be it out of distribution generalization or reducing sim to real gap.
- Few modules aren't easy to understand from the current version. For instance: the map dataset generation, association and feasibility of input text with the map in the map dataset.
- there are no details of the input format of map encoder, and human study involved.
- No qualitative results shown for other 2 baselines: MOTIONCLIP, TrafficGen

For language conditioned generation, While the current method doesn't seem to capture the infrastructure information like traffic signs, it would be interesting to see if the traffic light turns red, how does the behavior of the agents change at the intersection.

**Quality Of The Limitations Section:**

Limitations are addressed clearly

**Questions For Rebuttal:**

Need more details on the following to understand this work:

- How the map dataset is generated from waymo dataset?
- Most importantly, it is not clear how the input text is associated with the map. What if language says : "ego car overtakes a truck from the left neighboring lane", but if there  isn't any lane on the left of the ego vehicle in the map. Do you visualize each scenario
and then generate text corresponding to that based on feasibility of the scenario? How does it make sure that the structural representation obtained from GPT-4 is actually feasible with any map in the map dataset. And is the map sampled, to make sure that the scenario sampled from the dataset actually corresponds to the language description, before using MSE loss for prediction of agents' motion.
- How does the mapencoder work, does it take lane information as polylines, and learns graph adjacency matrix?
- Providing some statistics of the users involved in human study, for instance- their experience of driving, or their age, would be helpful.
- Further, Fig 4 and Fig 5 don't provide any qualitative results of the other 2 baselines: MotionCLIP and TrafficGen.
- I recently found this paper on similar domain, would be interesting to see how this methods is different, by including it in the related work [1].

[1]. https://arxiv.org/abs/2306.06344

**Robotics Focus:**

Highly relevant to robotics but no hardware experiments

**Summary Of Paper:**

The paper introduces LCTGen, a generative model combining LLMs and transformer based decoder to generate controllable and realistic traffic scenarios. It primarily consists of an interpreter and a generator, where interpreter converts the query to a structured representation of the scene consisting of map and agents scecific information using GPT-4. It follows a Chain of Thought promoting to summarize the scenario using GPT-4 and plan agent by agent to generate structured representation of the scene. Generator uses structured representation and sampled map to generate realistic traffic scenarios that matches the user's specifications. Results demonstrate that the proposed method outperforms the existing baselines (TrafficGen and MotionCLIP) for language conditioned generations and simulations, and is evaluated up with human studies.

**Summary Of Recommendation:**

I would be happy to increase my rating, if the issues are addressed in the updated draft.

---

### Official Review · Reviewer_ixPH · 2023-07-31

**Confidence:** 3
**Originality:** Good
**Technical Quality:** Good
**Clarity Of Presentation:** Good
**Impact:** 4

**Recommendation:**

Weak Accept: I recommend accepting the paper, but will not argue for my recommendation if the majority of other reviewers have a different opinion.

**Review:**

Strengths:
- The proposed approach enables use of real-world traffic incident datasets, which might be stored in textual format rather than geometric format. This can allow generation of simulated traffic scenarios that better reflect the type of incidents that actually occur in real life.
- The approach leverages the abilities of LLMs such as GPT-4 which excel at textual conversion tasks.
- The proposed traffic scenario generative model outperforms baselines.
- The human study shows that users find scenarios generated by LCTGen to better match the input text descriptions.

Weaknesses:
- The evaluation datasets for language-conditioned simulation appear to be small scale (40 cases for one, 38 cases for the other).
- The authors propose use of an LLM to interpret textual scenarios as a key component of their system. However, there appears to be no evaluation of that component in isolation.


**Quality Of The Limitations Section:**

Additional details required

**Questions For Rebuttal:**

The main issue I see is lack of evaluation of the LLM task. LLMs are known to ignore instructions, hallucinate facts in responses, or make mistakes. They may also make up reasons to justify what it outputted previously in order to stay consistent. As the use of the LLM is a key component of the proposed approach, it would be informative to evaluate how reliably the LLM actually accomplishes the text interpretation task.

Some ideas:
- Does the output (summary, explanation, and/or vectors) actually match what the input description says?
- Does the chain of thought have logical errors? Does the final answer match the intermediate outputs in the chain of thought?
- Does the LLM output actually follow all of the many guidelines specified?

Some other clarification questions:
- In the Waymo Open Dataset results, where does the language input L come from, if there is no language-scene paired data? Is the Interpreter not used in these experiments, or did someone write a description of each scenario?
- In baselines that don't use the Interpreter, where does the structured representation (the output of the Interpreter) come from?
- Intuitively, what kind of information does the language description add that allows LCTGen to perform better? Some analysis on this would be appreciated.

Minor comments:
- Sec. 4.1: 8-dimentional -> 8-dimensional
- Table 2: Ours Prefered -> Ours Preferred


**Robotics Focus:**

Relevant but unlikely to deploy to hardware in near future

**Summary Of Paper:**

This paper proposes a system called LCTGen for generating traffic scenarios from language descriptions. The system first uses an Interpreter module to convert a textual description of a traffic scenario into a compact, structured format. This is used by a Retrieval module to sample a map region that fits the structured scenario representation. The structured representation and map are given to a Generator module, which is a transformer model that outputs a scenario. The authors use the Waymo Open Dataset to evaluate scene reconstruction quality of the Generator module and find that it outperforms two baselines - TrafficGen and MotionCLIP. The authors then evaluate the language-conditioning aspect on two small datasets (NHTSA crash reports and Attribute Description dataset) using a human study. Finally, the authors show that LCTGen can also be used to edit traffic scenarios via commands expressed in natural language.

**Summary Of Recommendation:**

This paper proposes an interesting approach for language conditioned traffic scenario generation for testing self-driving in simulation. Experiments show strong performance of the proposed approach compared to baselines. However, additional analysis is needed for LLM task in the Interpreter module, as well as qualitative description of what information language adds that can explain the improved performance.

---

### Author Response · Authors · 2023-08-10
**Overall comment with LLM analysis**

We thank all the reviewers for their thoughtful reviews and helpful suggestions!

---

To better answer reviewers’ questions, we have uploaded a **pdf** file in each rebuttal. This file contains the following materials:
1. Output of LCTGen with four different LLMs (*Pages 1-9*), including generated scenarios by LCTGen and raw LLM outputs.
2. Qualitative results of LCTGen, MotionCLIP, and TrafficGen (*Pages 10-11*).
3. Statistics of our user study participants (*Page 12*).

---
As suggested by several reviewers, here we provide an analysis of LCTGen using different LLMs:

To study the effects of different LLMs, we provide the outputs of LCTGen using various state-of-the-art commercial and open-source LLMs, namely GPT-4, GPT-3.5, Llama2-7B [1], and Llama2-70B [1].

We provide four input texts from Crash Reports and Attribute Descriptions to each LLM. Then, we show the LCTGen output visualization in **Figures R1-R4** and the LLM output in **Texts 1-4** in our uploaded **rebuttal pdf**.

**Evaluation**:

For an intuitive comparison of the outputs from different LLMs, we evaluate different LLMs with the following criteria on these examples:
- Q1: Do the summary and explanation match the input text?
- Q2: Does the output vector match the input text?
- Q3: Does the LLM correctly use chain-of-thought (i.e., applies correct logic which matches the final answer)?
- Q4: Does the LLM follow the formatting guidelines?

We compute the average rate that these criteria are met for different LLMs:

|   | Q1 | Q2 | Q3 | Q4 |
| :--: | :--: | :--: | :--: | :--: |
| GPT-4 | 100% (1/1/1/1) | 100% (1/1/1/1) | 100% (1/1/1/1) | 100% (1/1/1/1) |
| GPT-3.5 | 100% (1/1/1/1) | 50% (0/1/1/0) | 75% (1/1/0/1) | 100% (1/1/1/1) |
| Llama2-7B | 75% (0/1/1/1) | 25% (1/0/0/0) | 50% (0/0/1/1) | 50% (0/0/1/1) |
| Llama2-70B | 25% (0/1/0/0) | 25% (1/0/0/0) | 0% (0/0/0/0) | 0% (0/0/0/0) |

This work was done manually within the scope of the rebuttal, and thus of limited scale. We will consider expanding this significantly for the final version of the paper.

**Observations**:

-  GPT-4 does well in chain-of-thought (CoT) inference and outputs vectors consistent with the CoT logic.
See an example on *Page 1 L9-17*, where the inference process matches well with the output. It also produces correct vectors most of the time.
-  GPT-3.5 sometimes makes commonsense mistakes.
 For example, on *Page 8 L33*, it says V1 turns right, however, on *L39* it indicates there is no intersection in the map. This commonsense error leads to the strange scenario in Figure R4.
- Llama 2 models often produce outputs that do not follow guidelines.
For example, on *Page 8 L36*, the model first outputs vectors and then outputs an explanation. This issue makes its chain-of-thought prompting ineffective and leads to incorrect results (e.g., Figure R2).
- Llama 2 models often make incorrect chain-of-thought inferences.
For example, on *Page 6 L54*, it indicates that V1 should turn left, but set V1’s action to 7 (turn right).
- Llama2-70B model refuses to answer two queries due to “safety concerns”.
For example, on *Page 9 L83*, it indicates the query may produce a “potential safety hazard” and refuse to answer. This is problematic as in the prompt we already indicate this is for traffic scenario generation, which is not harmful.

**Conclusion**:

- GPT-4 does well in terms of all the aspects most of the time, which aligns with the strong performance of LCTGen in our paper. We recommend using GPT-4 for LCTGen.
- GPT-3.5 works stably across different cases and produces reasonable outputs. However, it sometimes produces incorrect results with flawed chain-of-thought processes. One can use GPT-3.5 for LCTGen, but it might lead to inferior performance compared to GPT-4.
- Llama 2 models do not reliably work for LCTGen, they often output incorrect results and do not follow our guidelines (e.g., no chain-of-thought, refuse to generate). For the current version of LCTGen prompts, we do not recommend using Llama 2.

We might be able to engineer a prompt for Llama 2 and other open-source LLMs to work stably for LCTGen going forward.

[1] Llama 2: Open Foundation and Fine-Tuned Chat Models. Touvron et al.

---

### Decision · Program_Chairs · 2023-08-30

**Decision:**

Accept (Poster)

**Comment:**

The paper introduces LCTGen, a novel system for generating realistic traffic scenarios from natural language descriptions. It includes an Interpreter module for converting text into structured representations, a Generator module utilizing both structured input and map data to create coherent scenarios. The system outperforms existing benchmarks in language-conditioned scenario generation and scene quality, as demonstrated by evaluations on various datasets including Waymo Open Dataset and human studies. LCTGen's versatility extends to scenario editing through natural language commands and controllable driving policy evaluation, showcasing its potential for practical applications.

The reviewers pointed out the paper's strengths in its innovative utilization of LLMs for scenario generation, demonstrating performance improvements over baselines, and the exploration of novel applications such as language-conditioned editing. Additionally, the paper's high-quality writing contributes to its overall impact and accessibility.

The main weaknesses of the paper pointed out by the reviewers include that the evaluation datasets are small, the LLM component is not evaluated in isolation, and there is unclear or missing information in certain sections of the paper.

During the rebuttal phase, the authors provided informative new ablation study on different LLMs.

We recommend accepting this paper as poster. The authors should include the ablation study on different LLMs and address other weaknesses in the camera ready version.